# Calcified Neurocysticercosis: Demographic, Clinical, and Radiological Characteristics of a Large Hospital-Based Patient Cohort

**DOI:** 10.3390/pathogens13010026

**Published:** 2023-12-27

**Authors:** Javier A. Bustos, Gianfranco Arroyo, Oscar H. Del Brutto, Isidro Gonzales, Herbert Saavedra, Carolina Guzman, Sofia S. Sanchez-Boluarte, Kiran T. Thakur, Christina Coyle, Seth E. O’Neal, Hector H. Garcia

**Affiliations:** 1Center for Global Health, Universidad Peruana Cayetano Heredia, Lima 15202, Peru; javier.bustos.p@upch.pe (J.A.B.); carolina.guzman.d@upch.pe (C.G.); sssboluarte@gmail.com (S.S.S.-B.); hgarcia1@jhu.edu (H.H.G.); 2Cysticercosis Unit, Instituto Nacional de Ciencias Neurologicas, Lima 15030, Peru; isidrogonzalesq@hotmail.com (I.G.); hsaavedrapastor@hotmail.com (H.S.); oneals@ohsu.edu (S.E.O.); 3Direccion de Investigacion, Desarrollo e Innovacion, Universidad Cientifica del Sur, Lima 15067, Peru; 4School of Medicine and Research Center, Universidad Espiritu Santo-Ecuador, Samborondon 092301, Ecuador; oscardelbrutto@hotmail.com; 5Department of Neurology, Columbia University Irving Medical Center/New York Presbyterian Hospital, New York, NY 10032, USA; ktt2115@cumc.columbia.edu; 6Division of Infectious Diseases, Albert Einstein College of Medicine, Bronx, NY 10461, USA; christina.coyle@einsteinmed.edu; 7School of Public Health, Oregon Health & Science University-Portland State University, Portland, OR 97207, USA

**Keywords:** Neurocysticercosis, *Taenia solium*, calcification, epilepsy, Peru

## Abstract

Neurocysticercosis (NCC), the infection of the central nervous system caused by *Taenia solium* larvae (cysticerci), is a major cause of acquired epilepsy worldwide. Calcification in NCC is the most common neuroimaging finding among individuals with epilepsy in *T. solium*-endemic areas. We describe the demographic, clinical, and radiological profiles of a large hospital cohort of patients with calcified NCC in Peru (during the period 2012–2022) and compared profiles between patients with and without a previous known diagnosis of viable infection. A total of 524 patients were enrolled (mean age at enrollment: 40.2 ± 15.2 years, mean age at symptom onset: 29.1 ± 16.1 years, 56.3% women). Of those, 415 patients (79.2%) had previous seizures (median time with seizures: 5 years, interquartile range (IQR): 2–13 years; median number of seizures: 7 (IQR: 3–32)), of which 333 (80.2%) had predominantly focal to bilateral tonic-clonic seizures; and 358 (68.3%) used antiseizure medication). Patients had a median number of three calcifications (IQR: 1–7), mostly located in the frontal lobes (79%). In 282 patients (53.8%) there was a previous diagnosis of viable infection, while 242 only had evidence of calcified NCC since their initial neuroimaging. Most patients previously diagnosed with viable infection were male, had previous seizures, had seizures for a longer time, had more calcifications, and had a history of taeniasis more frequently than patients without previously diagnosed viable infection (all *p* < 0.05). Patients with calcified NCC were heterogeneous regarding burden of infection and clinical manifestations, and individuals who were diagnosed after parasites calcified presented with milder disease manifestations.

## 1. Introduction

Neurocysticercosis (NCC), a severe central nervous system (CNS) infection caused by the larval stage (cysticercus) of the tapeworm *Taenia solium,* is a major cause of acquired epilepsy worldwide. The infection and resulting disease are highly endemic in all parts of the developing world where pigs are raised in close contact with human feces [1,2]. NCC is endemic throughout Latin America, most of Asia including the Indian subcontinent, and extensive regions of China, Sub-Saharan Africa, and parts of Oceania [3,4,5,6,7,8]. NCC is also increasingly recognized in immigrant populations in upper- and middle-income countries due to an influx of immigrants seeking refugee status [9,10]. Despite its global prevalence and its negative economic impact [11,12], NCC remains a tropical neglected disease according to the World Health Organization (WHO) list [13].

Tapeworm larvae establish in the human brain initially as viable brain cysts causing only minimal or no inflammation [14,15]. However, at some point in time, either by natural evolution or as a result of antiparasitic treatment, a strong host inflammatory reaction leads to cyst degeneration and death [15,16,17]. Subsequently, some cysts disappear, whereas other become residual calcifications [18]. On computed tomography (CT), brain calcifications appear as small hyperdense lesions, with or without enhancement after the administration of contrast material [14]. Formerly believed to be an inert dead end of the parasite, calcifications persist in the host brain throughout life, remain as potential seizure foci, and may predispose hosts to a chronic epileptic state [18,19].

Up to one third of adult-onset seizures in *T. solium* endemic areas are attributable to NCC. Calcified NCC has been consistently associated with an increased risk of seizures and epilepsy in several studies [4,20,21]. Calcified lesions are the most frequent finding on brain CT in NCC cases from endemic areas (between 10% and 20%), although many infected individuals are presumed to be asymptomatic [22]. A large cross-sectional CT scan-based population study conducted in an endemic area, showed approximately three times the odds of having calcified NCC in people with epilepsy compared with those without epilepsy [23], whereas a prospective cohort study showed that the attributable fraction of incident adult-onset epilepsy due to NCC was approximately 30% [24]. On the other hand, most patients with symptomatic calcified NCC in hospital-based studies have seizures as the more frequent clinical manifestation, although headache, cognitive decline, and focal neurological signs can be seen with relative frequency [25,26,27]

The exact mechanisms leading to the development of epilepsy associated with calcified NCC are not fully understood. Pathological studies using human biopsies demonstrate perilesional gliosis and edema, abnormal vascular permeability, disruption of the blood-brain barrier (BBB), and axonal damage [28,29]. Also, the association between calcified NCC and hippocampal sclerosis (HAS) is increasingly reported, suggesting that both conditions may be interrelated and calcified NCC likely acts as the initial precipitating injury for the development of HAS [30,31,32].

Despite the high frequency of calcified NCC and its impact in neurological morbidity, clinical data is limited to case reports, small case series, or subgroups of patients in a series of individuals with NCC or individuals with epilepsy. There is also limited information on patient characteristics according to whether the disease was diagnosed when the infection was in its active form, or if symptoms appear only after parasite’s death. Therefore, this study presents the clinical, demographic, and radiological characteristics of a large cohort of patients with calcified NCC consecutively recruited in a referral neurological center in Peru, in order to provide a comprehensive description of the spectrum of disease associated with calcified NCC, the factors affecting its clinical presentation, and to identify if the profile of patients with previous diagnoses of active infection differs from those who present symptoms after lesions have been calcified.

## 2. Materials and Methods

### 2.1. Study Design

The NCCcal cohort is a prospective, longitudinal, hospital-based study designed to characterize the clinical course and factors associated with new symptomatic episodes in patients with calcified NCC of the brain parenchyma. This study incorporates patients attending the Cysticercosis Unit of the Instituto Nacional de Ciencias Neurologicas (INCN) in Lima, Peru for serological diagnosis of NCC. The INCN is a specialized, tertiary care-level neurological hospital considered the national referral center for neurological diseases. Patients who attended at the Cysticercosis Unit-INCN were referred from other hospitals and outpatient networks of neurological clinics in Lima and the provinces and included symptomatic or asymptomatic patients with lesions compatible with NCC on CT scan.

### 2.2. Participants, Enrollment, and Eligibility Criteria

From January 2012 to July 2022, patients aged 12 years and older with brain calcifications demonstrated on non-contrast CT (Figure 1) seen at the Cysticercosis Unit of the INCN were invited to participate in this study. A cutoff of 12 years was chosen since it has been shown that calcified NCC in disease-endemic communities affects children less often than adults [33,34]. Also, there is a high probability that seizures in children could be related to primary or idiopathic generalized episodes. We adapted the diagnostic criteria defined by Del Brutto (patients from endemic *T. solium* areas, with or without symptoms, and with lesions compatible with calcified NCC on CT) [35]. Calcified lesions were classified as NCC-related according to a typical appearance, pattern, and location. On CT scan, calcified lesions were identified as small, clearly demarcated, hyperdense rounded nodules or punctate lesions, with or without perilesional edema [36,37]. Patients received full information about the study through a detailed informed consent process by a study physician. Exclusion criteria included: presence of viable and/or degenerating CNS cysts located at the brain parenchyma, subarachnoid space, or within the ventricular system visualized on neuroimaging; a positive ratio >10 on Ag-ELISA (highly suggestive of viable infection); having primary generalized seizures; evidence of intracranial hypertension; or focal neurological deficits. In addition, pregnant women were not included in the study. Also, participants with calcifications in physiologic conditions such as the falx or choroid plexus were excluded from the calcified NCC group, as well as participants with calcification-patterns consistent with other known diseases such as Fahr disease and arterial atherosclerosis, or a vascular malformation [37].

### 2.3. Study Activities

Patients were interviewed by study physicians who were trained and supervised by senior neurologists of the INCN. In cases where patients reported uncontrolled seizures, they were referred to an epileptologist. Study physicians collected patients’ demographic and clinical information. Based on imaging reports and actual scans, two trained neurologists registered the number, size, characteristics, and topographic location of NCC calcifications visible on CT and ruled out the presence of viable and/or degenerating brain cysts on magnetic resonance imaging (MRI) or CT. Discrepancies between reviewers were resolved by consensus. Patients also underwent an electroencephalogram (EEG). The EEGs were standard, 30 min long, with hyperventilation and photic stimulation, using the international 10–20 system for electrode placement. Blood samples were also collected for antibody detection on enzyme-linked immunoelectrotransfer blot (EITB) assay [38] and antigen detection in B158/B60 Ag-ELISA [39]. 

### 2.4. Variables Investigated

Demographic variables included age at enrollment and sex. Clinical variables included the age at start of symptoms; the type of first symptom; having had previous seizures; years since first seizure; number of previous seizures; seizure-free time before enrollment in months; seizure semiology (seizures were classified as focal onset, with or without impaired awareness, and seizures that evolved to tonic-clonic bilateral seizures) [40]; history of status epilepticus; previous known diagnosis of viable NCC; previous neurosurgery; use of antiseizure medication (ASM); type of ASM, drug-resistant epilepsy; history of head trauma; perinatal complications; other CNS infections; previous diagnosis of taeniasis; family history of epilepsy; and NCC. EEG results were classified as normal or abnormal, and EITB results were reported as the number of antibody bands. According to brain CT, we defined the number of calcifications (continuous variable, single, from 2 to 10, and more than 10 calcifications), and their location in cerebral lobes (frontal, parietal, occipital, temporal or other).

### 2.5. Statistical Analyses

Data analysis was performed in Stata/SE 17.0 (Stata Corp LCC, College Station, TX, USA) and GraphPad Prism software, version 10.0.3. Demographic, clinical, and radiological characteristics of patients were described using summary statistics (frequencies and percentages for categorical variables, and mean ± standard deviation (SD) or median with interquartile ranges (IQR) for continuous variables). Missing information for each variable was also reported. Patient characteristics were compared between patients with and without previous diagnosis of viable NCC using bivariate analyses (chi-square test of independence for categorical variables, and Student’s *t*-test or Mann-Whitney U test for numerical variables according to the assessment of their normal distribution). Clinical characteristics were also compared between patients with single and multiple brain calcifications. A 5% significance level was considered for statistical significance. 

### 2.6. Ethics Statement and Protection of Human Subjects

This study protocol was approved by the Institutional Review Board (IRB) of the Universidad Peruana Cayetano Heredia (UPCH, approval number 69136), and the Instituto Nacional de Ciencias Neurologicas (INCN, approval number 014-2011), both in Lima, Peru. Participants were invited to be part of the study cohort and were enrolled after an informed consent process that included a written consent form. All exams that were part of the study were provided free of cost to the participants. 

## 3. Results

### 3.1. Study Flowchart

A total of 621 patients consecutively attending our unit were enrolled. Of these, 97 patients (15.6%) were excluded due to the following reasons: not having lesions compatible with calcified NCC on CT (*n* = 45); presence of viable brain cysts on neuroimaging (*n* = 9); positive Ag-ELISA (optical density ratio > 10 times the cutoff, *n* = 22); and missing information on brain neuroimaging and/or Ag-ELISA (*n* = 21; see flowchart in Figure 2). Therefore, 524 patients were included in the NCCcal cohort.

### 3.2. Demographic, Clinical, and Radiological Characteristics of Patients

The mean age (± SD) of patients at enrollment was 40.2 ± 15.2 years, and 295 (56.3%) were women. The mean age (± SD) of patients at the start of symptoms was 29.1 ± 16.1 years. When classified by the type of first symptom, most patients reported headache (273, 52.1%), followed by seizures (174, 33.2%), with 72 (13.7%) reporting seizures and headache, although headaches were not associated with seizure episodes. Of the 345 patients who reported headache as the first symptom, 126 received medical care, 60 received emergency care, and 18 were hospitalized. 

Overall, 415 patients (79.2%) had seizures at any point of their disease, with a median seizure time of five years (IQR: 2–13 years), and a median seizure-free time before enrollment of six months (IQR: 1–27 months). According to seizure type, most patients reported focal to bilateral tonic-clonic seizures (333/415, 80.2%). Additionally, 40/415 patients (9.6%) reported a history of status epilepticus, and only 2/415 patients (0.5%) had a history of febrile seizures.

A total of 353 out of 415 patients (85.1%) reported the use of ASM, with carbamazepine and phenytoin the most frequently used drugs (158/353, 44.8% and 131/353, 37.1%, respectively). Additionally, the median number of ASM drugs per patient was 2 (IQR: 1–3), and 35/353 patients (9.9%) who reported the use of ASM had drug-resistant epilepsy (Table 1).

A total of 228 patients (53.8%) had a previous diagnosis of viable NCC; 46 (8.8%) underwent surgical procedures for their disease. A total of 114 patients (21.8%) had a history of head trauma, of which 26 (22.8%) required emergency care but were not hospitalized. A total of 20 patients (3.8%) reported perinatal complications, and only 2 patients had other CNS infections (cerebral tuberculosis). A total of 71 patients (13.6%) reported a history of taeniasis, 130 (24.8%) had a family history of epilepsy, and 54 (10.3%) had a family history of NCC.

At enrollment, 370/524 patients (70.6%) had 1 or more EITB bands in serum, with a median number of 3 bands (IQR: 0–3 bands): 264/370 (71.4%) were positive for 1 to 3 bands, and 106/370 (28.6%) were positive for 4 or more bands. 

Seventeen patients (3.2%) had an abnormal baseline EEG result. Nine patients presented with cortical disfunction (four with generalized slowing and five with focal slowing), and six of them (75%) showed paroxysmal discharges, whereas only one patient with cortical disfunction showed interictal epileptic discharges. Six patients with abnormal EEG results presented with cortical-subcortical dysfunction, most of them focal or intermittent and with non-paroxysmal discharges. Only two patients had abnormal epileptiform activity (intermittent rhythmic theta activity in the EGG exam, see Appendix A). 

The median number of calcifications detected by CT per patient was 3 (IQR: 1–7 calcifications) with most of them having 2 to 10 calcifications (262, 50.0%), followed by 176 (33.6%) with single calcifications, and 86 (16.4%) with more than 10 calcifications. Calcifications were more often located in the frontal lobes (414, 79.0%), followed by the parietal lobes (242, 46.2%), the occipital lobes (181, 34.5%), and the temporal lobes (179, 34.2%).

### 3.3. Association between Patient Characteristics and Previous Diagnosis of Viable NCC

Patients with a known previous diagnosis of viable NCC had their initial symptoms at a younger age than those who did not (27.7 ± 14.9 years versus 30.7 ± 17.2 years, *p* = 0.038), and were most often male (141/282, 50.0% versus 88/242, 36.4%, *p* = 0.002, Figure 3A); had seizures more frequently (255/282, 90.4% versus 160/242, 66.1%, *p* < 0.001, Figure 3B); had seizures for a longer time (median: 6 years (IQR: 2–14) compared to median: 3 years (IQR: 1–11), *p* < 0.001, Figure 3C); had more seizure-free time before enrollment (median: 12 months (IQR: 2–34) compared to median: 2 months (IQR: 0–12), *p* < 0.001, Figure 3D); had more frequent neurosurgical procedures (39/282, 13.8% versus 7/242, 2.9%, *p* < 0.001); had a history of more frequent taeniasis (39/282, 13.8% versus 7/242, 2.9%, *p* < 0.001, Figure 3E); had stronger antibody reactions on EITB (median: 3 bands (IQR: 2–5) versus median: 1 band (IQR: 0–3), *p* < 0.001); had more brain calcifications (median: 3 calcifications (IQR: 1–9) versus median: 2 calcifications (IQR: 1–5), *p* < 0.001; and 61/282 patients, 21.6% had more than 10 calcifications versus 25/242 patients with no previous diagnosis of viable NCC, 10.3%, *p* < 0.001, Figure 3F). The proportion of patients with abnormal EEGs was higher for those without a previous diagnosis of viable NCC (12/233, 5.2%, versus 5/276, 1.8%, *p* = 0.037, Appendix A).

### 3.4. Clinical Characteristics of Patients with Single or Multiple Brain Calcifications

A history of previous seizures was more frequent in patients with multiple brain calcifications compared to patients with single calcifications (286/348, 82.2% and 129/176, 73.3%, *p* = 0.018). Among patients with previous seizures, the median time from first seizure to enrollment was also longer in patients with multiple calcifications (median: 5 years (IQR: 2–14) compared to median: 3 years (IQR: 1–10) in patients with single calcified lesions, *p* = 0.047). A trend towards a longer seizure-free time before enrollment was also observed in patients with multiple calcifications (median: 7 months (IQR: 1–31) compared to median: 4 months (IQR: 0–20) in patients with single calcifications, *p* = 0.071). The remaining clinical variables were not statistically different between patients with single or multiple brain calcifications (Appendix A).

## 4. Discussion

Brain calcifications play an important role in seizure burden and neurological morbidity in NCC patients [17,18]. This study reports the baseline demographic, clinical, and radiological profiles of a large, hospital-based cohort of patients with calcified NCC: 524 patients seen at a referral neurological center in Peru. Nearly 80% of the enrolled patients with calcified NCC had previous seizures, and more than half of them also had a previous diagnosis of viable NCC. Furthermore, patient profiles differed between patients with and without a previous diagnosis of viable NCC. The data provided here represent a close approximation to the spectrum of neurological morbidity associated with calcified NCC and should serve to orientate physicians treating patients with this type of NCC.

The overall mean age and predominance of women enrolled in the NCCcal cohort were similar to those previously reported in other hospital-based studies [41,42]. The exact reason for the higher proportion of women among patients with calcified NCC is unknown, although it has been suggested that women elicit a more aggressive inflammatory reaction towards brain cysts, increasing the likelihood of developing residual calcifications [43,44]. Approximately 80% of patients with calcified NCC enrolled in this cohort reported a history of previous seizures, which ranged between 50% and 80% as previously reported in other studies of symptomatic NCC cases [14,25]. These percentages markedly differ from population-based studies in which more NCC cases were asymptomatic [22]. Most patients with a history of previous seizures enrolled in the NCCcal cohort reported focal to bilateral tonic-clonic seizures as the predominant type (80.2%), similar to previous studies in NCC patients [45], whereas pure focal seizures were reported in a low proportion. The reason for this can reflect a memory bias effect causing participants to report more aggressive crises and not recognize low-intensity seizures, although we cannot exclude other factors such as genetic predisposition; cyst load in the CNS; and antiparasitic treatment with consequent neuroinflammation, which can also have an effect on triggering bilateral tonic-clonic seizures. 

More than 80% of patients with calcified NCC who reported seizures also used ASM at enrollment, which contrasts with low rates of ASM use among people with NCC in population-based studies in endemic areas [46] and likely represents a referral bias considering the specialized nature of our institution. The most commonly used ASMs reported by our patients were carbamazepine, phenytoin, and valproic acid. These drugs are known for their more noticeable side effects and drug interactions compared to other second-generation anticonvulsant drugs. This may affect patient adherence to treatment during follow-up and underlines the need to optimize ASM therapy.

Less than 35% of patients in our cohort had single calcifications, which contrasts with the higher frequency of cases with single calcifications in the Indian subcontinent (approximately 80%) [47,48]. This can be explained by differences in the epidemiological characteristics of *T. solium* infection in India compared to Latin America. In India the majority of the population is vegetarian; few individuals raise pigs which leads to lower numbers of tapeworm carriers, so infections may occur by indirect transmission through dispersal mechanisms such as contaminated water or food [48,49].

An interesting finding was the high frequency of patients in the cohort with headache as the presenting clinical feature. In the majority of studies, the presence of seizures as the most frequent present clinical feature is high [25]. Therefore, it is possible that our findings represent patients’ recall bias towards headache rather than seizures (generally of low intensity at the beginning). The association between the presence of calcified NCC and primary headache has also been suggested, likely as a consequence of the remodeling process of calcification and the release of antigen remnants in the brain parenchyma [50]. 

Individuals with calcified NCC are not a homogeneous population. Compared with patients who presented with already calcified NCC on neuroimaging, patients with a previous diagnosis of viable NCC presented a clearly different clinical profile. They were younger at symptom onset (consistent with the natural evolution of intraparenchymal cysts) [51,52]; had seizures in a higher proportion; reported a history of taeniasis more frequently; had more calcifications; and had a longer seizure-free time before enrollment. Seizures in this subgroup of patients may have occurred when the cyst was viable or had been provoked by neuroinflammation in earlier stages of cyst degeneration [53,54], including transient periods of neuroinflammation induced by the use of antiparasitic treatment [15]. There are reports of patients with taeniasis and who also had severe forms of NCC (e.g., multiple parenchymal or extraparenchymal cysts) [55,56], suggesting secondary autoinfection due to exposure to a large number of eggs. The higher number of brain calcifications in patients with a previous diagnosis of viable NCC may also suggest higher initial cyst loads, affecting the prognosis of a seizure disorder [41]. It is also likely that a higher proportion of patients with previous viable NCC received antiparasitic treatment, which may lead to increased risk of seizures and other neurological symptoms during therapy [15]. 

A longer seizure-free time before enrollment may result from better ASM adherence, and subsequently good seizure control, in individuals who passed through the experience of a diagnosis of viable brain cysts and antiparasitic therapy, compared to individuals diagnosed with already calcified parasites and debuting with seizures later in life.

The main strength of our study was the evaluation of a large hospital cohort of patients with calcified NCC, which allowed a better characterization of the clinical spectrum of the disease. Likewise, as the INCN is a national reference center for neurological diseases, our results reflected the characteristics of the hospital population from different areas of Peru including some *T. solium* endemic areas. We also demonstrated different clinical profiles between cases attending at the INCN with a previous diagnosis of viable infection, and those who attend for care only after lesions calcified. These findings can serve to guide the physician for the appropriate therapeutic approach in patients. 

Our study has some drawbacks. First, there was a potential recall bias among patients with seizures and calcified NCC, as living with this chronic condition may have affected the information about seizures they reported at enrollment. Second, patients’ clinical information related to seizures was obtained using non-standard definitions, making it difficult to compare our findings with other studies. Third, we based our definition of calcified NCC primarily on CT scans provided by patients. The use of CT in resource-limited settings such as in our study is more affordable for the diagnosis of NCC, and for cases with calcified NCC, a CT scan is highly sensitive and correlates with the calcification burden [57,58]. Fourth, due to financial resource limitations, we were not able to perform an MRI exam on all patients at enrollment in order to rule out the presence of viable brain cysts (due to a sensitivity close to 100%) [59,60]. It is likely that some patients enrolled in the NCCcal cohort had viable cysts that were not identified by brain CT. Nonetheless, the presence of high positive ratios (>10) on Ag-ELISA as an exclusion criterion should have significantly reduced the probability of enrolling patients with viable cysts missed by CT. It is also likely that in the group of patients with bilateral tonic-clonic seizures we also included some patients with unknown onset-motor to bilateral tonic-clonic seizures [48]. In addition, we did not evaluate additional information on other psychiatric comorbidities or quality of life of patients at enrollment. This information is being prospectively evaluated and will be presented in future studies. Finally, we only included patients who attended at the INCN, the national reference center for neurological diseases, although a large percentage of people with calcified NCC in the community live without symptoms or knowledge about their disease. Patients with calcified NCC in our cohort are self-selected by being neurologically symptomatic (approximately 80%) and as such their representativeness relates to this population rather than the overall population of individuals in endemic communities.

## 5. Conclusions

This large clinical cohort of patients with calcified NCC provides a comprehensive view of the spectrum of neurological disease associated with this late stage of the disease and demonstrates that individuals with calcified NCC are a quite heterogeneous population. The data we provide here should guide future studies to demonstrate the real burden of neurological disease and unveil the underlying pathogenic mechanisms causing brain inflammation and damage in calcified NCC. 

## Figures and Tables

**Figure 1 pathogens-13-00026-f001:**
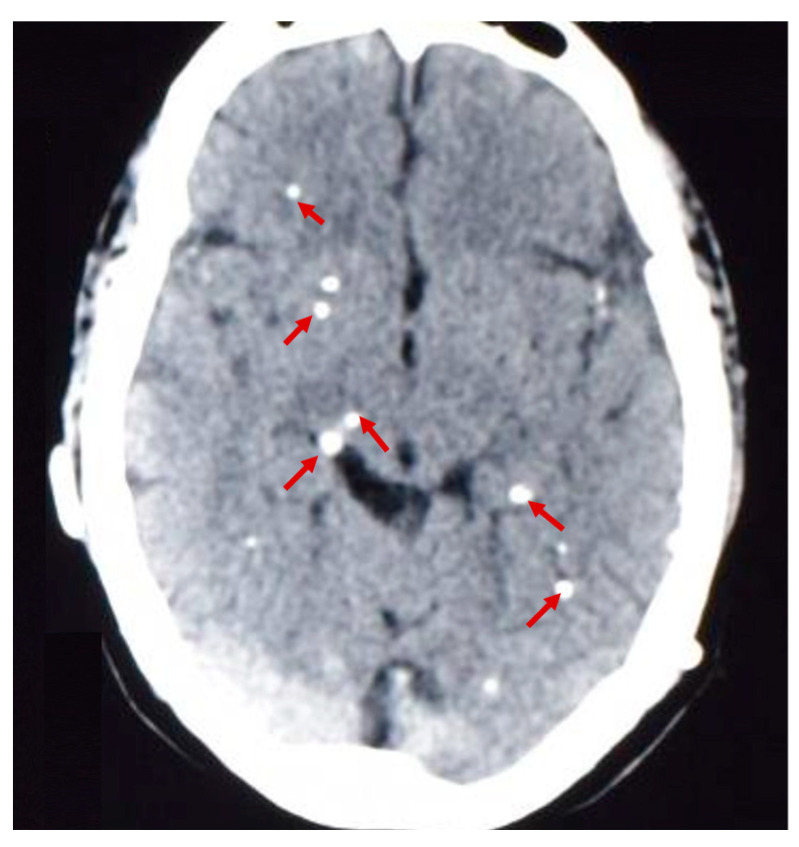
Calcified lesions (red arrows) seen in the brain parenchyma of a patient with NCC on non-contrast CT.

**Figure 2 pathogens-13-00026-f002:**
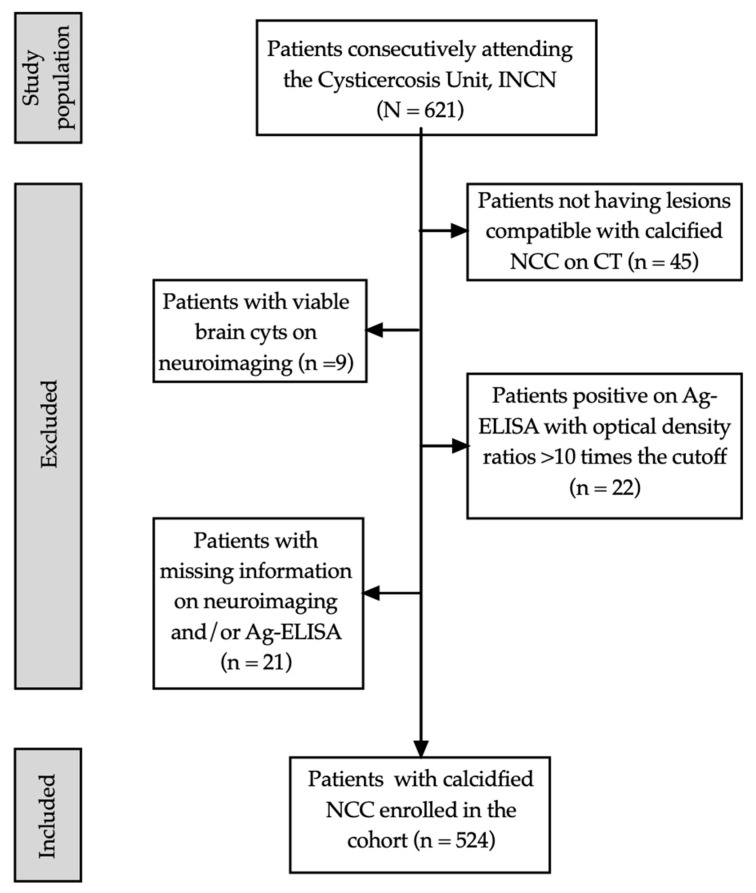
Flowchart of study population selection.

**Figure 3 pathogens-13-00026-f003:**
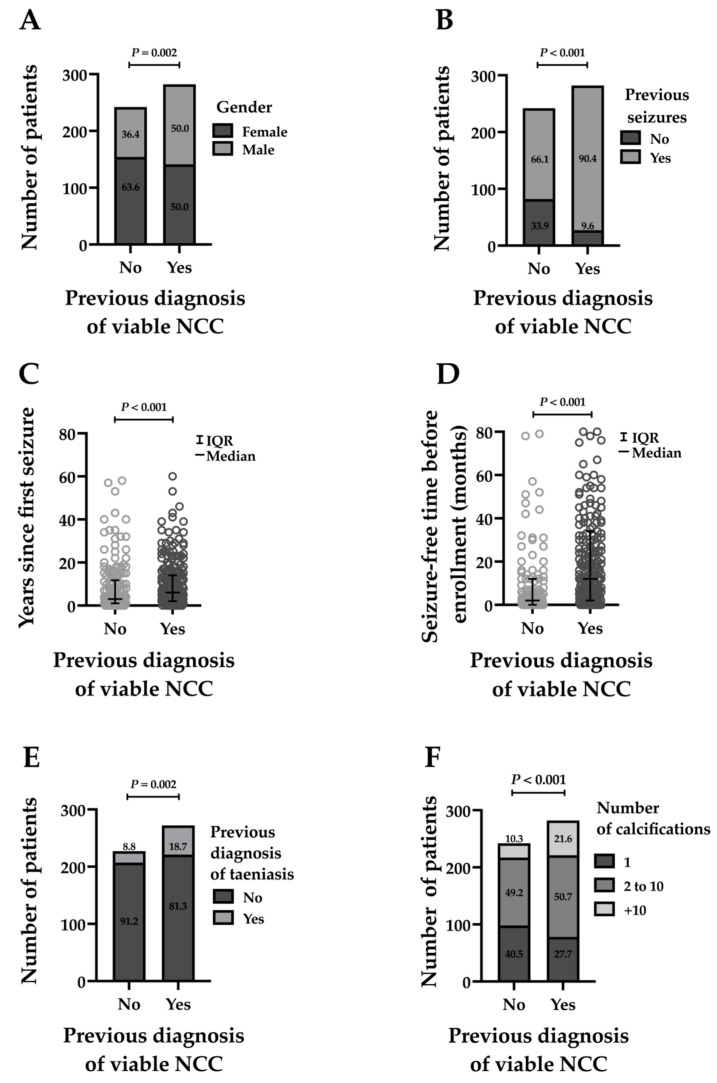
Significant association between previous diagnosis of viable NCC with gender (**A**), history of previous seizures (**B**), years since first seizure (**C**), seizure-free time before enrollment in months (**D**), previous diagnosis of taeniasis (**E**), and number of calcifications (**F**).

**Table 1 pathogens-13-00026-t001:** Demographic, clinical, and radiological characteristics of patients with calcified NCC enrolled in the cohort (N = 524).

Characteristics	*n* (%)
Age at enrollment (years) Mean ± SD	40.2 ± 15.2
*Sex* Female Male	295 (56.3) 229 (43.7)
Age at start of symptoms (years) Mean ± SD	29.1 ± 16.1
Type of first symptom No symptoms Headache Seizures Headache and seizures	1 (0.2) 273 (52.1) 174 (33.2) 72 (13.7)
Previous seizure No Yes	109 (20.8) 415 (79.2)
Years since first seizure † Median (IQR)	5 (2–13)
Number of previous seizures †, ‡ Median (IQR)	7 (3–32)
Seizure semiology † Pure focal seizure Focal to bilateral tonic-clonic seizures	82 (19.8) 333 (80.2)
History of status epilepticus † No Yes Not described/missing data	373 (89.9) 40 (9.6) 2 (0.5)
ASM † No Yes Not described/missing data	52 (12.5) 353 (85.1) 10 (2.4)
Type of ASM ¶ Carbamazepine Phenytoin Valproic acid Clobazam Other Not described/missing data	158 (44.8) 131 (37.1) 46 (13.0) 16 (4.5) 1 (0.3) 1 (0.3)
Number of ASM drugs ¶ Median (IQR)	2 (1–3)
Drug-resistant epilepsy ¶ No Yes	318 (90.1) 35 (9.9)
Seizure-free time before enrollment (months) †^,^* Median (IQR)	6 (1–27)
Previous diagnosis of viable NCC No Yes	242 (46.2) 282 (53.8)
Previous neurosurgery No Yes	478 (91.2) 46 (8.8)
Head trauma No Yes Not described/missing data	404 (77.1) 114 (21.8) 6 (1.1)
Perinatal complications No Yes Not described/missing	498 (95.0) 20 (3.8) 6 (1.2)
Previous diagnosis of taeniasis No Yes Not described/missing data	428 (81.7) 71 (13.5) 25 (4.8)
Family history of seizures No Yes Not described/missing data	369 (70.4) 130 (24.8) 25 (4.8)
Family history of NCC No Yes Not described/missing data	445 (84.9) 54 (10.3) 25 (4.8)
Baseline (at entrance to cohort) EEG results Normal Abnormal Not described/missing data	492 (93.9) 17 (3.2) 15 (2.9)
EITB results (number of antibody bands) Median (IQR)	3 (0–3)
Number of calcifications on brain CT Median (IQR) 1 2 to 10 +10	3 (1–7) 176 (33.6) 262 (50.0) 86 (16.4)
Topographic location of calcifications in cerebral lobes Frontal Parietal Occipital Temporal	414 (79.0) 242 (46.2) 181 (34.5) 179 (34.2)

Abbreviations: NCC (neurocysticercosis); SD (standard deviation); IQR (interquartile range); EEG (electroencephalogram); EITB (enzyme-linked immunoelectrotransfer blot); ASM (anti-seizure medication); CT (computed tomography); † evaluated in patients with previous seizure (*n* = 415); ‡ five missing observations; ¶ evaluated in patients that reported use of ASM (*n* = 353); * one missing observation.

## Data Availability

The data presented in this study are available on request from the corresponding author. The data are not publicly available due to participant privacy.

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
