# Peer review of "Calcified Neurocysticercosis: Demographic, Clinical, and Radiological Characteristics of a Large Hospital-Based Patient Cohort"

_pathogens, 2023, doi:10.3390/pathogens13010026_

Round 1

Reviewer 1 Report

Comments and Suggestions for Authors

Firstly, I would like to congratulate the authors on their relevant and interesting work.

However, some points need to be addressed:

1)      In the last paragraph of the introduction, it would interesting if the authors specified the research question, what were the initial hypotheses of the authors for this study, including what does this study add that is new to the published literature.

2)      Please include information about the specific study protocol that was chosen. Was it previously validated?

3)      Please specify how were inclusion and exclusion criteria chosen, including the age cutoff

4)      Please provide more information about the setting and specify what were the criteria for referral to the “Cysticercosis Unit”(are asymptomatic patients with NCC lesions seen in this clinic, are patients referrals from the ER, from outpatient setting, self)

5)      Please inform time elapsed from the initial symptomatic presentation until the treatment

6)      Were any specific radiological diagnostic criteria for CT used in this study?

7)      Considering the limitations of head CT, please specify why was MRI not used in all of the patients and expand the “limitations” section on the discussion including epidemiological data comparing sensitivity and specificity for NCC detection in CT vs MRI.

8)      Was the power of the study calculated?

9)      Was the distribution of the variables normal?

10)  Did any of the patients present drug-resistant epilepsy?

11)  Were any of the patients using more than one ASM?

12)  Please provide more information about EEGs, for how long were patients monitored.

13)  Please provide further information about semiology of headaches, including frequency and intensity, in the patients who were classified as having headaches and seizures, did headaches occur before, during or after seizures

14)  Please provide further information about seizure frequency, duration, subtypes of seizures in semiological classification, presence or absence of auras and prodromes

15)  In any of the patients, was there failure of first ASM?

16)  Did any of the patients have febrile seizures?

17)  Did any of the patients had a history of head trauma?

18)  Did any of the patients have history of perinatal complications or other CNS infections?

19)  Were there any psychiatric comorbidities in these patients?

20)   Were patients with seizures also seen by epileptologists?

21)  Was video-EEG done in any of the patients?

22)  Was quality of life assessed in any of the patients?

23)  Please provide further explanation in the discussion about what the study adds new to the literature and its external validity.

Comments on the Quality of English Language

Minor editing for typo correction.

Author Response

REVIEWER 1:

Question 1: In the last paragraph of the introduction, it would interesting if the authors specified the research question, what were the initial hypotheses of the authors for this study, including what this study adds that is new to the published literature

Response: It has been modified as suggested and it now reads “Despite the high frequency of calcified NCC and its impact in neurological morbidity, clinical data is limited to case reports, small case series, or subgroups of patients in series of individuals with NCC or individuals with epilepsy. There is also limited in-formation on patients’ characteristics according to whether the disease was diagnosed when the infection was in its active form or if symptoms appear only after parasite’s death. Therefore, this study presents the clinical, demographic, and radiological characteristics of a large cohort of patients with calcified NCC consecutively recruited in a referral neurological center in Peru to provide a comprehensive description of the spectrum of disease associated with calcified NCC, the factors affecting its clinical presentation, and to identify if the profile of patients with previous diagnosis of active infection differs from those who present symptoms after lesions have been calcified”.

Question 2: Please include information about the specific study protocol that was chosen. ¿Was it previously validated?

Response: For the diagnosis of NCC we followed the diagnostic criteria of Del Brutto in addition to the classic morphology of parenchymal calcifications as referred now in 2.2 Participants enrollment and eligibility criteria “We adapted the diagnostic criteria defined by Del Brutto (patients from endemic T. solium areas, with or without symptoms, and with lesions compatible with calcified NCC on CT) [35]. Calcified lesions were classified as NCC-related according to a typical appearance, pattern, and location. On CT scan, calcified lesions were identified as small, clearly demarcated hyperdense rounded nodules or punctate lesions, with or without perilesional edema”.  Seizures and epilepsy were identified and classified according to the ILAE 2017 recommendation. It has been included in 2.4 Variables investigated “Clinical variables included the age at start of symptoms, the type of first symptom, having had previous seizures, years since first seizure, number of previous seizures, seizure-free time before enrollment in months, seizure semiology (seizures were classified as focal onset, with or without impaired awareness, and seizures that evolved to tonic-clonic bilateral seizures) [40]

Question 3: Please specify how the inclusion and exclusion criteria were chosen, including the age cutoff.

Response: This information is included in the modified version of the manuscript, and it now readsA cutoff of 12 years was considered for enrollment since calcified NCC in disease-endemic communities have shown that children are less often affected than adults. There is also a high probability that seizures in children could be related to primary or idiopathic generalized episodes”.

Question 4. Please provide more information about the setting and specify what were the criteria for referral to the “Cysticercosis Unit” (are asymptomatic patients with NCC lesions seen in this clinic, are patients’ referrals from the ER, from outpatient setting, self)

Response: This information has been included in the modified version of the manuscript, and it now reads “This study incorporates patients attending at Cysticercosis Unit of the Instituto Nacional de Ciencias Neurologicas (INCN) in Lima, Peru for serological diagnosis of NCC. The INCN is a specialized, tertiary care level neurological hospital and is considered the national referral center for neurological diseases. Patients who attended at the Cysticercosis Unit - INCN were referred from other hospitals and outpatient networks of neurological clinics in Lima and the provinces and included symptomatic or asymptomatic patients with lesions compatible with NCC on CT scan”.

Question 5: Please inform time elapsed from the initial symptomatic presentation until the treatment.

Response: We appreciate this observation. Unfortunately, during the patients’ interview, we focused mainly on the previous diagnosis of viable NCC. Previous antiparasitic treatment could also have been referred by the patient for any other parasitic infection (hydatidosis, fasciolosis, etc.). However, this information will be obtained prospectively during patient follow-up.

Question 6: ¿Were any specific radiological diagnostic criteria for CT used in this study?

Response: This information has been included in the modified version of the manuscript, and it now reads “We adapted the diagnostic criteria defined by Del Brutto (patients from endemic T. solium areas, with or without symptoms, and with lesions compatible with calcified NCC on CT scan. Calcified lesions were classified as calcified NCC-related according to a typical appearance, pattern, and location. On CT scan, calcified lesions were identified as small, clearly demarcated hyperdense rounded nodules or punctate lesions, with or without perilesional edema”.

Question 7: Considering the limitations of head CT, please specify ¿why was MRI not used in all of the patients? and expand the “limitations” section on the discussion including epidemiological data comparing sensitivity and specificity for NCC detection in CT vs MRI.

Response: This information has been added in the discussion section of the modified manuscript, and it now reads “Third, we based our definition of calcified NCC primarily on CT scans provided by patients. The use of CT in resource-limited settings such as in our study is more affordable for the diagnosis of NCC, and for cases with calcified NCC CT scan is highly sensitive and correlates with the calcification burden [56,57]. Fourth, due to financial resource limitations, we were not able to perform an MRI exam to all patients at enrollment to rule out the presence of viable brain cysts (due to its sensitivity close to 100%) [58,59]. It is likely that some patients enrolled in the NCCcal cohort had viable cysts that were not identified by brain CT. Nonetheless, the presence of high positive ratios (>10) on Ag-ELISA as an exclusion criterion should have significantly reduced the probability of enrolling patients with viable cysts missed by CT

Question 8: ¿Was the power of the study calculated?

Response: Power was not calculated for this study as it was a consecutive series of patients with calcified NCC enrolled in a cohort.

Question 9: Was the distribution of the variables normal?

Response: It has been included in the modified manuscript and it now reads “Patients’ characteristics were compared between patients with and without previous diagnosis of active NCC infection using bivariate analyses (Chi square test of independence for categorical variables, and Student’s T test or Mann-Whitney U test for numerical variables according to the assessment of their normal distribution)”.

Question 10: ¿Did any of the patients present drug-resistant epilepsy?

Response: We included this variable in the results section, and it now reads “Additionally, the median number of ASM drugs per patient was 2 (IQR: 1-3), and 35/353 patients (9.9%) who reported the use of ASM had drug-resistant epilepsy (Table 1)”.

Question 11: ¿Were any of the patients using more than one ASM?

Response: We included the variable number of ASM drugs used per patient in the results section, and it now reads “Additionally, the median number of ASM drugs per patient was 2 (IQR: 1-3).

Question 12: Please provide information about EEGs, ¿for how long patients were monitored?

Response: This information is included in the modified manuscript, and it now reads “Patients also underwent an electroencephalogram (EEG). The EEGs were standard, 30 minutes long, with hyperventilation and photic stimulation, using the international 10-20 system for electrode placement”.

Question 13: Please provide further information about semiology of headaches, including frequency and intensity, in the patients who were classified as having headaches and seizures, did headaches occur before, during or after seizures

Response: Semiology regarding headaches was classified into those that required either outpatient or emergency medical care, emergency care, or hospitalization. Of 345 people who reported the symptom, 126 received some type of medical care, 60 received emergency care, and 18 were hospitalized. This information is included in the results section, and headaches were not associated with seizure episodes.

Question 14: Please provide further information about seizure frequency, duration, subtypes of seizures in semiological classification, presence, or absence of auras and prodromes

Response: This information is described in the results section. Specific information regarding auras and prodromes has not been collected at enrollment but is being collected during the follow-up of patients.

Question 15: In any of the patients ¿Was there failure to first ASM?

            Response: We only collected information regarding drug resistance, for those cases who continue presenting seizure after having used two or more than 2 regimens of antiepileptic drugs.

Question 16: ¿Did any of the patients have febrile seizures?

Response: This information has been included in the modified manuscript, and it now reads “Additionally, 40/415 patients (9.6%) reported a history of status epilepticus, and only 2/415 patients (0.5%) had a history of febrile seizures”.

Question 17. ¿Did any of the patients have a history of head trauma?

Response: This information has been included in the modified manuscript, and it now reads “Two hundred twenty-eight patients (53.8%) had a previous diagnosis of viable NCC, 46 (8.8%) underwent surgical procedures for their disease, 114 patients (21.8%) had a history of head trauma, of which, 26 (22.8%) require emergency, but were not hospitalized”.

Question 18: ¿Did any of the patients have a history of perinatal complications or other CNS infections?

Response: This information has been included in the modified manuscript, and it now reads “Twenty patients (3.8%) reported perinatal complications, and only two patients had other CNS infections (cerebral tuberculosis)”.

Question 19: ¿Were there any psychiatric comorbidities in these patients?

Response: We did not collect this information at enrollment. We mentioned this in the discussion section of the modified manuscript as follow “Also, we did not evaluate additional information on other psychiatric comorbidities or quality of life of patients at enrollment. This information is being prospectively evaluated and will be presented in future studies”.

Question 20: ¿Were patients with seizures also seen by epileptologists?

Response: This information has been included, and it now reads “Patients were interviewed by study physicians who were trained and supervised by senior neurologists of the INCN. In cases where patients reported uncontrolled seizures, they were referred to an epileptologist”.

Question 21: ¿Was video-EEG done in any of the patients?

Response: No. Only standard EEG exams were conducted in patients at enrollment.

Question 22: ¿Was quality of life assessed in any of the patients?

Response: We did not collect this information at enrollment. We mentioned this in the discussion section of the modified manuscript, and it now reads “Also, we did not evaluate additional information on other psychiatric comorbidities or quality of life of patients at enrollment. This information is being prospectively evaluated and will be presented in future studies”.

Question 23: Please provide further explanation in the discussion about what the study adds new to the literature and its external validity

Response: This information has been added in the modified manuscript, and it now reads “The main strength of our study was the evaluation of a large hospital cohort of patients with calcified NCC, which allows a better characterization of the clinical spectrum of the disease. Likewise, being the INCN a national reference center for neurological dis-eases, our results reflect the characteristics of the hospital population from different areas of Peru including some T. solium endemic areas. We also demonstrated different clinical profiles between cases attending at the INCN with a previous diagnosis of viable infection and those who attend for care only after lesions calcified. These findings can serve to guide the physician for the appropriate therapeutic approach in patients”.

Reviewer 2 Report

Comments and Suggestions for Authors

The mentioned study is precisely processed and the results are clearly interpreted. The study reports interesting correlations between individual monitored parameters in patients with calcified Neurocysticercosis.

Author Response

.-

Reviewer 3 Report

Comments and Suggestions for Authors

Comments on the Quality of English Language

Nil

Author Response

Reviewer #3:

Question 24: In the third paragraph I suggest that the authors should add data from the following their own studies

  • Del Brutto OH, Arroyo G, Del Brutto et al. Epilepsia 2017, 58, 1955–2017.
  • Del Brutto OH, Recalde BY, Mera RM. Am. J. Trop. Med. Hyg. 2022, 106, 208–214

Response: These references have been included in the modified manuscript.

Question 25: The authors mentioned patients with primary generalized seizures as one of the exclusion criteria. The term primary generalized seizure presently refers to GTCS in patients with idiopathic epilepsy, JME, JAE, and epilepsy with GTCS only. Patients with NCC can present with GTCS and also motor seizures with unknown onset to bilateral tonic and clonic. Can the authors clarify this point?

Response: Although unlikely, it is possible that some calcified-NCC patients presented primary motor seizures of unknown origin. These patients could have been classified in the group of cases with bilateral tonic-clonic seizures. This information is included in the limitation section of the discussion.

Question 26: Seizure semiology (seizures were categorized as either pure focal seizures or focal to bilateral tonic-clonic seizures). It will be appropriate to say, “focal onset with or without impaired awareness”. The appropriate references should be added.

Response: This information and the reference cite has been included in the modified manuscript, and it now reads “seizure semiology (seizures were classified as focal onset, with or without impaired awareness, and seizures that evolved to tonic-clonic bilateral seizures)”.

Question 27: The important observations of the study were most patients previously diagnosed with viable infection were males, had previous seizures, had seizures for a longer time, had more calcifications, and had a history of taeniasis more frequently compared to patients without previously diagnosed viable infection (all P<0.05). Even the authors had discussed some possible reasons for the differences, it needs a comprehensive discussion.

Response: This information has been modified and it now reads: “Individuals with calcified NCC are not a homogeneous population. Compared with patients who presented with already calcified NCC on neuroimaging, patients with a previous diagnosis of viable NCC presented a clearly different clinical profile. They were younger at symptom onset (consistent with the natural evolution of intraparenchymal cysts) [50,51], had seizures in a higher proportion, reported a history of taeniasis more frequently, had more calcifications, and had a longerseizure-free time before enrollment. Seizures in this subgroup of patients may have occurred when the cyst was viable or had been provoked by neuroinflammation in earlier stages of cyst degeneration [52,53], including transient periods of inflammation induced by the use of antiparasitic treatment.[15] There are reports of patients with taeniasis and who also had severe forms of NCC (e.g., multiple parenchymal or extraparenchymal cysts) [54,55], suggesting some sort of autoinfection and higher or sustained exposure to tapeworm eggs. The higher number of brain calcifications in patients with previous diagnosis of viable NCC may also suggest higher initial cyst loads, affecting the prognosis of the seizure disorder [40]. All considered, these differences point to a subgroup of individuals who have longer disease evolution and a heavier lesional burden”.

Question 28: Single calcific lesion is seen in 176 (33.6%) patients in this study, whereas the reported frequency of solitary calcific NCC in India is about 80%, can the authors discuss the possible reasons. I also strongly suggest that the authors should compare clinical characteristics of patients with solitary calcific lesion with the clinical characteristics of patients with multiple calcific lesions.

Response: All this information has been included in the modified manuscript (results and discussion section), and it now reads:

“Results: A history of previous seizures was more frequent in patients with multiple brain calcifications compared to patients with single calcifications (286/348 [82.2%] and 129//176 [73.3%], P = 0.018). Among patients with previous seizures, the median time from first seizure to enrollment was also longer in patients with multiple calcifications (median: 5 years [IQR: 2-14] compared to median: 3 years [IQR: 1-10] in patients with single calcified lesions, P = 0.047). A trend towards a longer seizure-free time before enrollment was also observed in patients with multiple calcifications (median: 7 months [IQR: 1-31] compared to median: 4 months [IQR: 0-20] in patients with single calcifications, P = 0.071). The remaining clinical variables were not statistically different between patients with single or multiple brain calcifications (Supplementary Table S3)”.

“Discussion: Less than 35% of patients in our cohort had single calcifications, which contrast with the higher frequency of cases with single calcifications in the Indian subcontinent (ap-proximately 80%) [46,47]. This can be explained by the differences in the epidemiological characteristics of T. solium infection in India compared to Latin America, since in India the majority of the population is vegetarian, few individuals raise pigs leaving to few numbers of tapeworm carriers, so infections may occur by indirect transmission through dispersal mechanisms such as contaminated water or food [47,48]. 

Question 29: Clinical feature which needs some explanation is the high frequency of headache as the presenting feature (52.1%) and the frequency of seizures as the presenting feature is only 33.1%, In most of the reported series the frequency of seizures as the presenting feature is high. A short discussion may be added in the discussion.

Response: This information is now included in the modified manuscript, and it now reads “An interesting finding was the high frequency of patients in the cohort with headache as the presenting clinical feature. In the majority of studies, the presence of seizures as the most frequent present clinical feature is high [25]. Therefore, it is possible that our findings represent patients’ recall bias towards headache than seizures (generally of low intensity at the beginning). The association between the presence of calcified NCC and primary headache has also been suggested, likely as a consequence of the remodeling process of calcification and the release of antigen remnants in the brain parenchyma [49]”.

Round 2

Reviewer 3 Report

Comments and Suggestions for Authors

I am satisfied with the revised manuscript. 

Comments on the Quality of English Language

minor